# A rescued dataset of sub-daily meteorological observations for Europe and the southern Mediterranean region, 1877–2012

5

Linden Ashcroft<sup>1,2,</sup> Joan Ramon Coll<sup>1</sup>, Alba Gilabert<sup>1</sup>, Peter Domonkos<sup>1</sup>, Enric Aguilar<sup>1</sup>, Javier Sigro<sup>1</sup>, Mercè Castella<sup>1</sup>, Per Unden<sup>3</sup>, Ian Harris<sup>4</sup>, Phil Jones<sup>4,5</sup> and Manola Brunet<sup>1,4</sup>

<sup>1</sup>Centre for Climate Change, Department of Geography, Universitat Rovira i Virgili, Tarragona, Spain
 <sup>2</sup>Australian Bureau of Meteorology, Melbourne, Australia
 <sup>3</sup>Swedish Meteorological and Hydrological Institute, Folkborgsvägen, Norrköping, Sweden
 <sup>4</sup>Climate Research Unit, School of Environmental Sciences, University of East Anglia, Norwich, United Kingdom

<sup>5</sup>Center of Excellence for Climate Change Research, Department of Meteorology, King Abdulaziz University, Jeddah, Saudi Arabia

Correspondence to: Linden Ashcroft (linden.ashcroft@bom.gov.au)

**Abstract.** Sub-daily meteorological observations are needed for input to and assessment of high-resolution reanalysis products to improve understanding of weather and climate variability. While there are millions such weather observations that have been collected by various organizations, many are yet to be transcribed into a useable format. Under the auspices of the European Union funded Uncertainties in Ensembles of Regional ReAnalysis (UERRA)

- project, we describe the compilation and development of a digital dataset of 8.8 million meteorological observations rescued across the European and southern Mediterranean region, many of them Essential Climate Variables (ECVs) as defined by the Global Climate Observing System (GCOS). By presenting the entire chain of data preparation, from the identification of regions lacking in digitized sub-daily data and the locating of original sources, through the digitization of the observations to the quality control procedures applied, we provide a rescued dataset that is as traceable as
- possible for use by the research community. Data from 127 stations and of 15 climate variables in the northern Africa and European sectors have been prepared for the period 1877 to 2012. Quality control of the data using a two-step semi-automatic statistical approach identified 3.5 % of observations that required correction or removal, on par with previous data rescue efforts. In addition to providing a new sub-daily meteorological dataset for the research community, our experience in the
- development of this UERRA sub-daily dataset gives us an opportunity to share guidance on future data rescue projects.
   All data are available on PANGAEA: <u>https://doi.pangaea.de/10.1594/PANGAEA.886511</u>.
   Keywords: meteorological data, Europe, Mediterranean, essential climate variables, data rescue

#### 1. Introduction

Digitizing meteorological observations into a useable modern format is crucial for long-term climate monitoring and assessment. High-quality, long-term observations are needed for almost all aspects of meteorological and climatological research, but many spatial and temporal gaps still exist in data products currently used by the international research

- 5 community (Brunet and Jones, 2011). For this reason, meteorological data rescue and recovery is becoming increasingly important, particularly in developing countries and for the early instrumental period, as data are often only available in paper format and are at great risk of being permanently lost (Brunet and Jones, 2011; Page et al., 2004; World Meteorological Organization, 2016).
- In the last 20 years, many initiatives have been established to recover and digitize land-based meteorological observations at national, regional, and international scales. The Atmospheric Circulation Reconstructions over the Earth initiative (ACRE, Allan *et al.*, 2011) coordinates data rescue across the globe, while other projects such as MEditerranean DAta REscue (MEDARE, www.omm.urv.cat/MEDARE/index.html) and Historical Instrumental Climatological Surface Time Series Of The Greater Alpine Region (HISTALP, www.zamg.ac.at/histalp/) focus on particular regions (Auer et al., 2007; Brunet et al., 2014a, 2014b). Additional initiatives on a national to regional scale,
- led by meteorological agencies (e.g. Kaspar *et al.*, 2015 in Germany) and research projects (e.g. Ashcroft *et al.*, 2014; Brunet *et al.*, 2006, 2014a) have located and digitized historic observations, and ensured that they are made available to the scientific community.

Many of these projects have focused on the rescue of daily, monthly and or annually-averaged data, as these observations form the basis of long-term climate analysis. Daily maximum temperature, minimum temperature and

- precipitation totals are often the top priority for digitization, because these variables are used to monitor changes in climate and the incidences of extreme weather events, as well as to identify economic and agriculturally important long-term variations in precipitation (Brunet et al., 2006; Moberg et al., 2006). The development of the 20<sup>th</sup> Century Reanalysis product which uses only sub-daily atmospheric pressure observations as input for a global reanalysis has also benefited from national and regional data rescue activities, resulting in an increase in atmospheric pressure data
- recovery in recent years (Compo et al., 2011; Cram et al., 2015). Far fewer recovery efforts have been made to uncover sub-daily meteorological observations of other variables, despite the fact that they are the necessary input to global and regional reanalysis products, the output of which can greatly improve the understanding of atmospheric circulation and of high-temporal resolution extreme events (e.g. Cannon *et al.*, 2015; Stickler *et al.*, 2014).

This paper presents the experience and resultant dataset of a two-year digitization effort aimed at recovering sub-daily

- meteorological data. Our work formed part of Uncertainties in Ensembles of Regional ReAnalysis (UERRA, http://uerra.eu/), a project under the European Union 7th Framework Programme. The goal of UERRA was to produce ensembles of European regional reanalyses at high temporal resolution for several decades, with an estimate of the associated uncertainties in the resulting datasets. A key component of UERRA is the recovery of sub-daily surface meteorological observations to provide input to and assess the quality of the various reanalysis products.
- In this paper we describe our complete data rescue process to provide sufficient details, as much as possible, for a fully traceable dataset. In Sect. 2 we explain how we identified target regions for data rescue across Europe and the neighboring southern Mediterranean region to maximise improvement in spatial and temporal coverage of existing data, as well as potential sources of sub-daily data for digitization. We present the methods employed to minimise errors in the digitization process and the steps required to take the data from a disparate set of sources to a unified database.

In Sect. 3 we provide details on the quality assurance and control procedures used to reduce errors in the dataset, including visual checks, semi-automatic statistical methods and an automatic spatial comparison method. We present the dataset and quality control results in Sect. 4. Finally, we give details about how to access the data, as well as some practical ideas for future data recovery projects, based on our experiences with this particular project.

### 5 **2.** Methods and materials

#### 2.1. Identifying gaps in sub-daily data availability

The primary goal of the data rescue efforts within UERRA has been to improve spatial and temporal coverage of input data for future regional gridded and reanalysis climate products over the European domain. Adding new observations to the available network of datasets from which these products are derived will help to reduce uncertainties, ultimately
helping to improve the understanding of European weather and climate. This involves, as a first step, identifying the basic station data used in current reanalysis products available at the European Centre for Medium Weather Forecasts (ECMWF) and other relevant databases that contain digitized observations.

To identify gaps in the available sub-daily climate record, we first conducted an extensive examination of the Meteorological Archival and Retrieval System (MARS) of the ECMWF. MARS is home to the primary data input of

15 the current European reanalysis products available from ECMWF, and so stations that are identified in data sources (see Sect. 2.2) but not present in MARS, or stations with low percentages of sub-daily data are likely candidates for data recovery. We divided our search into pre-1950 and post-1950 data availability, to align with the temporal focus of the proposed UERRA regional reanalysis products and ECMWF historical reanalyses such as ERA-20C (https://www.ecmwf.int/en/research/climate-reanalysis/era-20c). The variables of interest were several Essential

- Climate Variables (ECVs) as defined by the Global Climate Observing System (GCOS) as well as two other variables, deemed useful for reanalysis by the UERRA research team. The ECVs we focussed on were air temperature (TT), atmospheric pressure (sea level pressure, PP and station level pressure, SP), wind speed (WS) and wind direction (WD), relative humidity (RH), dew point temperature (DP) and daily rainfall (RR). The non-ECVs were snow depth (SD) and fresh snowfall (FS).
- Next, we examined other sub-daily data repositories where rescued observations are likely to be stored to further minimise potential duplication of data digitization efforts. We cross-referenced our MARS results with the station list of the International Surface Pressure Databank (ISPD, Cram *et al.*, 2015), the Koninklijk Nederlands Meteorologisch Instituut (KNMI) European Climate Assessment and Dataset (ECA&D: <u>http://eca.knmi.nl/</u>), and the national climate data systems of countries whose data may not yet be in a multi-national repository. In particular, we examined the data
- available from the National Climate Data Management Systems (CDMS) of the Romanian Meteorological Administration (NMA-RO) and the National Meteorological and Hydrological Services (NMHS) of countries in the Western Balkans, including Albania; Bosnia & Herzegovina; the Republic of Macedonia; Montenegro; and the Republic of Serbia.

With this data availability information, we were able to identify the Mediterranean, Eastern Europe and Scandinavia as
 three key sub-regions within the European sector where MARS and other data repositories were lacking in sub-daily data for reanalysis development, particularly in the post-1950 period (Fig. 1, Table S1).

The high percentage of stations with data for less than 20 % of the 1950–2010 period (Fig. 1) illustrates the lack of subdaily observations in these sectors. Gaps are clear in the southern and eastern Mediterranean countries, Sweden and

Norway for the 1960s and 1970s (Table S1), as well as across the Balkan region. The relatively dense spatial coverage of the stations also suggests there is a high likelihood that observations have been taken at many places in these regions, but have not yet been made available in a standardised format.

#### 2.2. Locating and assessing scans of sub-daily data sources

- As well as identifying gaps in the digitized sub-daily record available for Europe, we also needed to locate sources of undigitized sub-daily data. We undertook extensive consultation with NMHS across the three identified regions of poor data coverage, in an attempt to identify and recover paper or scanned data sources suitable for digitization. Priorities were given to data sources already available as scanned images, and stations with variables identified as important for the development and verification of regional reanalyses (see Sect. 2.1). Recovered precipitation observations from
- NMA-RO were digitized internally, and then provided to us in digitized quality controlled format, using a similar quality control format to ours (see Sect. 3.2). Discussion with the Norway and Swedish NMHS uncovered data for these countries that had been digitized, but were not yet provided to international data repositories. Similarly, the Catalan Meteorological Service (MeteoCat), which has an open data policy, offered us their digitized data for the recent period 1998–2015 to be transferred to relevant global repositories through our effort. Data sharing was organised
- between these regions and ECMWF without the need for observations to be transcribed from paper format, and will therefore not be discussed in the current study. Political and financial difficulties prevented many countries we contacted, particularly in northern Africa and the Balkans regions, from providing original data sources to us for digitization.

Original data sources were provided in scanned format by Deutscher Wetterdienst (the German Meteorological Office,

DWD), the Slovenian Environmental Agency (SEA), and Agencia Estatal de Meteorología (the Spanish Meteorological Service, AEMET), via MeteoCat. Close consultation with these NMHS enabled us to identify valuable and previously undigitized data sources. From these sources, stations with minimal data available in MARS were selected for digitization.

The World Meteorological Organization (WMO) MEDARE initiative and the precursor project to UERRA, the EU-

25 funded EURO4M project (http://www.euro4m.eu/), located key records of data for the Middle Eastern, Balkan and southern Mediterranean regions from the Serbian NMHS online climatological scanned repository (http://www.hidmet.gov.rs/ciril/

meteorologija/klimatologija\_godisnjaci.php), the United States of America's National Oceanic and Atmospheric Administration/National Climatic Data Center (NOAA/NCDC) Climate Data Modernization Project (CDMP:

- <u>http://library.noaa.gov/Collections/Digital-Documents/Foreign-Climate-Data-Home</u>), the British Atmospheric Data Centre (BADC, <u>http://badc.nerc.ac.uk/browse/badc/corral/images/metobs</u>) and other national meteorological services (see Brunet *et al.*, 2014a, 2014b for details). Daily maximum and minimum temperature, precipitation, and sub-daily atmospheric air pressure observations from some of these sources were digitized under the auspices of EURO4M and MEDARE, but many other observations were unable to be transcribed due to project constraints. UERRA therefore
- provides a valuable opportunity to rescue the previously undigitized values from these sources (Brunet et al., 2014b). Table 1 provides detail of the data sources identified for digitization, while Fig. 2 shows several examples of the data sources used. All of the variables included in each source are listed in Table 1, although not all were digitized under the auspices of UERRA. The majority of data sources from CDMP are secondary, meaning that they are collations or summaries of observations that have been prepared in a central location. Unfortunately, secondary data sources are
- more prone to transcription errors than original series, as they have been transferred from the original readings. Many were handwritten, although a small subset was typed.

### 2.3. Digitizing method

Once the data sources had been identified and catalogued, a group of 11 digitizers were employed 15 hours a week over a two-year period to digitize the data. The digitization team was made up of undergraduate and postgraduate geography students from the University Rovira i Virgili, who all had some knowledge of meteorological variables and European climate. The digitizers worked on desktop computers in a computer lab, with large screens and standard keyboards.

They were also given the option of working from home on their personal laptops.

The digitizers received initial training sessions, as well as online instructions and monthly in-person meetings to discuss issues and introduce new digitization tasks. Digitization was done using a strict "key as you see" method, meaning that the digitizers typed the values that were provided in the data images, rather than using any coding system. This follows standard best practise outlined by the WMO (2016).

Budget constraints made it unfeasible to employ double-keying, a suggested method of improving digitized data quality (Brönnimann et al., 2006). We tested optical character recognition (OCR) and speech recognition technologies, but the diverse nature of each task and the time and cost associated with training the software for each data source made these options unfeasible. However, the digitizers were trained in self-assessment techniques aimed at reducing data errors.

Digitizers were asked to carefully cross check their values with the original source values for the 10th, 20th and 30th day of each month to make sure that no days had been skipped or repeated. Days with missing data were recorded in metadata files, along with any other variations in the data source. Where data sources included monthly totals and summaries, digitizers were instructed to calculate these values from their daily transcribed data, to check accuracy.

The data sources were in a number of different formats (see Fig. 2). The two main formats were one month (or day) to a 20 page for a single station, and one day to a page for a network of stations. Depending on the source structure, each digitizer was in charge of digitizing values from a station (e.g. Egyptian and Moroccan sources, Fig. 2a and b), a time period (e.g. Slovenia, Fig. 2c), or a variable (e.g. Lebanon, Fig. 2d).

In several cases, not all of the data on a sheet were required to be digitized, as they had already been transcribed as part of EURO4M and MEDARE. To help digitizers with the complex layout of the source images, templates were developed in Microsoft Excel for some sources that were as close as possible to the format of the original data source (e.g. Fig. 3). Borders and shading within the files were used to help the digitizer keep track of their work, and date columns were pre-filled with the correct dates to reduce the occurrence of errors associated with leap years. The development of templates was not always possible due to time constraints, although it was employed for all sources

with hourly data (see Table 1).
The digitizers were required to upload their data to a central server every 15 days, include a count of the number of values digitized, and a copy of the data transcribed so far. This method ensured that the digitizers were making progress, the data were being regularly backed up, and that the digitized observations could be regularly checked (see Sect. 3).

#### 2.4. Conversion to standard units

35 While all quality control and assessment was applied to the data in their original units, the data were also converted to standard units, to be used in widespread meteorological products and statistical quality control procedures (Table 2). Data sources and available metadata were examined closely to ensure the conversions were as accurate as possible, and any changes to units within the same source were captured. Many atmospheric pressure observations needed to be

converted from mm of mercury to hectopascals, and station level pressure data reduced to sea level pressure for quality control testing. This step involved a detailed examination of the data sources to identify station height information and any instrument movements that may have occurred.

#### 3. Quality assessment of digitized data

5 Quality control (QC) procedures are crucial to identify non-systematic errors that could be hidden in time series. These errors can occur as a result of issues with original sources, the method of data collection, transcription, or the digitization process. Ensuring that data are digitized with the utmost consideration for data reliability, and applying a QC procedure to the digitized observations are essential steps in the preparation and analysis of climate data. This is particularly the case for daily or sub-daily data, as these observations are used in the calculation of monthly and annual 10 means.

An ideal QC procedure must be transparent and rigorous to ensure internal data consistency, temporal and spatial coherence, and traceability for future data users. A well-defined and executed QC routine will be able to flag data errors from time series that could compromise the analysis of natural climate variability and anthropogenic climate change, including the study of extreme variables. This is the key to avoid incorrect climate interpretations induced by data errors

15 in a climate change context (Aguilar et al., 2003; Brunet et al., 2006)

An exhaustive QC application was vital for our study, but given the large number of observations, completely manual QC was not a feasible method of correcting the data. However, a completely automated procedure, such as that used for global databases (Dunn et al., 2012) would also be sub-optimal, as the digitized data do not cover a wide geographic area and consistent time period. We therefore decided that a multiple-step process would be the best approach.

Figure 4 outlines the multiple steps of the data quality assurance and control procedures used in the development of the dataset. As outlined in Sect. 2, efforts were made before digitization to minimise the introduction of errors, including a detailed assessment of each data source, the development of templates for many sources, and the selection of qualified digitizers. During and after digitization, the digitized data were then subjected to quality control and assurance testing. The structure of the testing (Fig. 4) can be summarised as a basic visual check, statistical testing at the individual station level, and spatial testing across comparable networks.

Note that homogenization is not included in this procedure. Although the homogenization of data to remove nonclimatic features of a long-term instrumental record is crucial for the assessment of climate variability and change (e.g. Peterson *et al.*, 1998), homogeneity assessment of sub-daily data is a highly complex task that is still in development within the research community (Venema et al., 2012).

### 30 3.1. Visual cross-checking

Values uploaded by digitizers were systematically compared to the original source images by climatologists familiar with the sources, and occasionally other digitizers. The aims of these initial visual crosschecks was to provide timely feedback to the digitizers if common errors were occurring, identify subtle errors in the order of the data that may not be picked up in statistical procedures, and also make a preliminary assessment of the quality of the data from each

35 particular source (Table 1). Additionally, regular reporting of data completed helped us identify any digitizers who were having trouble with their tasks and needed extra assistance.

10

25

30

For every fourth year of data, two or three days of observations were selected at three monthly intervals for visual cross checking with the original source. Additional ad hoc checks were made if a known issue existed in the data source e.g. if the period covered by the data source contained a leap year, or the source pages were known to be out of order. Although these checks only covered a small percentage of the total digitized data, we felt it was sufficient to identify the

5 general quality of work done by individual digitizers and for each source.

In more than 60 % of stations tested, only a small number (less than 5 %) of the checked values required correction. Visual cross checking of data from stations with a larger number of errors identified the occasional skipped day or duplicated value, which meant that a large percentage of observations needed to be shifted by one time step. The majority of these errors were found in data for Egypt and Algeria, from sources that had already been flagged as difficult to read and containing date order errors. In two cases, digitizers were asked to repeat their work.

#### 3.2. Individual station quality control (SAQC method)

After the basic visual quality checks, the digitized data were subjected to a range of statistical quality control tests. Due to the highly variable nature of the different data sources, and their disparate geographical spread, data from each station were examined individually in this step. Statistical quality control was conducted using a semi-automatic quality

15 control (SAQC) procedure developed by the Centre for Climate Change at URV (http://www.c3.urv.cat/softdata.php). SAQC comprised of three separate programs that can be applied to data in text file format: one examining temperature, wind, relative humidity and dewpoint observations; another assessing sea level pressure data, and a final check on sub-daily rainfall data, daily snow depth and snow fall. The tests were largely adapted from existing automatic quality control procedures developed for sub-daily data at a global scale (e.g. Dunn *et al.*, 2012; Durre *et al.*, 2010), but were 20 adapted for the UERRA dataset to enable more manual examination of the resultant flags. The tests applied within

SAQC (Table 3) can be largely grouped into four groups depending on the degree of QC applied (Aguilar et al., 2003):

- *Gross errors tests*: QC tests that detect and flag obviously erroneous values (date order check, date errors, unrealistic values, data repetitions and non-numeric value tests).
- Tolerance tests: QC tests that detect and flag those values considered outliers with respect to their own-defined upper/lower limits (climatic outliers, bivariate comparisons, monthly mean of absolute increments, and unusual distribution of values tests).
- Internal consistency check: QC tests which detect and flag incoherencies between associated elements within each record (Interval and DP/FS/SD inconsistency test, RH/DP/TT comparison tests, precipitation and snow totals test)
- *Temporal coherency*: QC tests which detect and flag a given value that is not consistent with the amount of change that might be expected in a variable in any time interval according to adjacent values (Flat line test, big jump test, summer snow test and irregular temporal evolution).

Each program was applied at a country level, producing a list of values flagged by each test at each station. The results of each test were then manually cross referenced against the original source data, and corrected or removed by a trained climatologist. The removal or correction of each value was recorded using a flag system, to clearly document the nature

of the identified errors and results (Table 4). An example of the air temperature evolution in Port Said (Egypt) taken at 0800 and 1400 for the short period 1939–1940 and resultant QC flags is shown in Fig. 5, highlighting various types of errors, outliers and extreme values over a short time period.

In the initial testing of the SAQC procedure, the tests for duplicate values, monthly mean of absolute increments and unusual distribution of values tests were found to be overly sensitive, resulting in many valid observations being

flagged for assessment. Many of the legitimate errors identified by these tests were also found by others, so the thresholds on these tests were relaxed to make the task of checking flagged values more manageable.

#### **3.3.** Spatial quality assurance (HQC method)

- The final QC procedure consisted of subjecting data from neighboring stations to spatial quality control tests. Only data that had been checked by visual and automatic QC were subjected to this procedure. The spatial QC process was conducted using an adapted version of the procedure used in the development of the U.K. Met Office Hadley Centre Global Sub-Daily Station Observations dataset (HadISD v2.0.1.2016p; Dunn et al., 2012, 2016). Adaptation of some tests (Table 5) was required as the UERRA dataset had low spatial resolution and included observations taken at inconsistent times, often converted from units with coarse resolution. Automatically running HQC in its standard form
- 10 led to a large number of false positive flags being identified, automatically removing a significant number of correct observations being removed from the dataset (Fig. 6).

To reduce the number of false positive flags, the minimum number of neighboring stations required for HQC testing was reduced from ten to five, and the percentage of non-missing observations per month allowed was reduced from 75 % to 66 %. Tests that looked for streaks of identical values, or non-uniform distributions in the frequency of values were also slackened to account for the fact that many observations were converted from different units.

- The 127 stations were then split into networks according to their correlation, spatial distance, observing times, overlapping observing periods and variables observed. Six appropriate networks were identified (Table 6), but unfortunately it was not possible to include all stations, periods, variables and observing times. The heterogeneous characteristics of the dataset, the high spatial distance and low spatial resolution of the stations and the inconsistent
- coverage of the variables included in the dataset meant that only about 4.3 million observations (over 48 % of the total dataset) could be subjected to HQC.

For example, it was not possible to apply HQC to data from Cyprus, Lebanon and Spain due to the low number of stations in each country and the large spatial distance from the neighboring country stations. We were also unable to analyse fresh snow and snow depth, precipitation or relative humidity data, as the HadISD QC does not assess these

25 variables. Moreover, several stations (such as those in Germany and Slovenia, network 6 in Table 6) provided hourly data, but there were not enough neighboring stations with sufficiently high temporal resolution to allow for more than several observing times per day to be checked.

#### 3.4. Final check

After the HQC was applied, a final check was made to ensure that the conversion procedures had been applied correctly 30 and that all flags were realistic i.e. that a flag of 3 was associated with a value that had been removed from the final datasets.

#### 4. Results

#### 4.1. Spatial and temporal data distribution

15

A total of 8.8 million observations were digitized from 127 stations in 15 countries (Table 7 and Table S2). The majority of located sources provided sub-daily temperature observations, wind speed and wind direction, and atmospheric pressure. A small network of sources from Germany, Lebanon and Slovenia contained hourly data for a number of variables, contributing to the high number of observations for those countries. Additional sources from

30

Slovenia contained sub-daily rainfall data, while sources provided by the SEA and DWD included daily snowfall and snow depth data from Slovenia and Germany.

Long records (> 30 years) with many variables were successfully recovered from stations in Egypt, Tunisia and Algeria, although only the Egyptian stations provide observations more than once a day (Fig. 7). Shorter but more widespread

- observations were rescued across Morocco, Turkey and in the Balkans region, while the snowfall observations in Germany only covered the west of the country. The largest number of observations (more than 28 %) came from Slovenia (Fig. 8a); even though we only had data for three stations in Slovenia, the observations were hourly, included nine variables, and covered more than 20 years. Around 15 % of the rescued observations came from Egypt, and almost 12 % from Turkey. Both of these countries have
- a large number of stations in the recovered network, and a variety of variables over a long period of time (Fig. 7). More than 21 % (1.8 million) of the rescued observations were sub-daily temperature measurements, with wind speed and direction measurements totalling over 17 % (Fig. 8b). There were around 20,000 more wind direction observations than wind speed; this is because very early Tunisian and Egyptian wind speed observations were qualitative (e.g. light, moderate) and were not digitized. Relative humidity data made up around 16 % of the rescued dataset, while sea level
- pressure and station level pressure contributed a similar amount at just over 15 % (around 1.4 million values). Over 160,000 fresh snow and 160,000 snow depth values (more than 3.5 % combined) were also recovered from Germany and Slovenia from as far back as the 1950s, representing a significant increase in snow observations across the region. Due to the temporal coverage of the Slovenian data (1950–1978), as well as the dedicated focus of the UERRA project on post-1950 observations, the mid-20<sup>th</sup> Century was the most well represented period in the rescued dataset (Fig. 8c).
- Almost 60 % of the dataset covered the 20 years from 1950 to 1969. Observations from Cyprus and northern Africa provided data from the late 19<sup>th</sup> century, and records from Serbia were recovered up to 2012. Finally, the most common observing times for the variables rescued were 0700, 1400 and 2100, reflecting standard observing practises over the European region in the 20<sup>th</sup> century. Tunisian observations were only available for 0700, and for many other countries where observations were only available once a day in the early part of the record, these
- observations were inevitably in the morning also. Two German stations included a small number of half hourly observations (Fig. 8d).

#### 4.2. Semi-automatic quality control results (SAQC)

All rescued sub-daily data were subjected to quality control routines to identify erroneous values or chains of values in the time series (Sect. 3). A total of 3.2 % of observations, around 268,000, were flagged as suspicious for the whole UERRA dataset using SAQC (Fig. 9).

Removing correct values that have been flagged (false positives) is a common QC issue, and manual examination ensured that these important observations – often of extreme events – are retained for future studies. The majority of the values flagged (1.5 % of the total number of values) were corrected after manual examination, with just over 1 % of the total number of observations deleted due to errors in the source image or issues with the readability of the original

values. Over 27, 000, 0.3 % of the total number of observations, were flagged but then found to be correct after examination.

Despite being the country with the smallest number of observations, the largest percentages of flagged values found were for Bosnia and Herzegovina and the Czech Republic (~8 % of the total number of data digitized, Fig. 10a). For Bosnia and Herzegovina a large section of observations from one station needed to be set to missing due to digitizer

error, and for the Czech Republic observations a digitization error was able to be corrected by shifting data by one day. The hand-written nature of the Czech data, together with the absence of data templates (only used in Slovenian, Spanish

and German data sources) may go some way to explaining the large number of flagged values among both countries. The countries with the largest number of observations (Egypt and Slovenia) had about 3 % of their observations corrected or verified, and less than 2 % removed under the SAQC procedure.

A similar amount of flagged values were proportionally found in all rescued observations distributed by variables,

- except for precipitation (RR, Fig.10b). The high number of precipitation flags is due to two factors. Firstly, several 5 digitizers inadvertently recorded zero rainfall values as missing, or missing rainfall as zero. The format of the Slovenian data sources changed over the period, with some years having hourly rainfall data and others only providing observations three or four times a day. This issue can significantly skew any resulting analysis and so was corrected wherever it was identified.
- 10 Secondly, during the latter part of the Slovenian record, some daily rainfall totals were calculated inconsistently, using a midnight to midnight sum occasionally rather than a 0700-0700 total. The six-hourly observations were QCed based on these totals, but the daily rainfall totals calculated in this way were removed from the final version of the dataset to ensure consistency.
- SAQC flags distributed by decade show a similar pattern to the distribution of observations, with a peak in the mid-20th 15 century (Fig.10c). The higher number of fl17 flags (observations set to missing as no value could be found in the source image) during the 1940s may reflect data issues during the Second World War, particularly for Egypt and Algeria, where there were issues associated with the ordering of the original source files. Flagged values were relatively evenly distributed across observation times (Fig.10d), although the lower absolute numbers of half hourly observations made for a higher proportion of flagged observations proportionally found in all rescued observations distributed by
- observation times. 20

#### 4.3. Spatial quality control results (HQC) Quality control results

Temperature was the variable with the smallest number of flagged values overall by HQC, with the exception of network 2 where data source resolution and the high percentage of temporal gaps lead to extra flags (Fig. 11). The variable with the highest proportion of flagged values in network 2 was sea level pressure.

- Given the automatic nature of the HQC tests, all values flagged by this step were removed and given a flag of 36. 25 Values that were subjected to HQC were therefore marked with an additional flag (a prefix of 3), to clearly identify the level of testing applied to each individual observation (see Table 5 and Fig. 12). This means that observations which were corrected or verified in the SAQC round of testing (and given a flag of 2 or 4) but passed the spatial QC procedure had a final flag of fl32 or fl34, ensuring that information from both rounds of QC were retained to maintain the traceability of the QC procedure. 30
  - In total about 64,000 values were flagged and subsequently removed by HQC, around 0.7 % of the total dataset (Fig. 12). While the HQC tests were unable to be applied to all of the observations, these results are similar to the findings of other large-scale QC efforts (Dunn et al., 2012). Around 3.9 %, or about 330,000 observations were flagged by both QC procedures (Fig. 12). A total of 2.1 % of the data were removed as a result of SAQC and HQC testing, with 1.5 %
- 35 corrected during the SAQC process. Only 0.3 % were flagged but later verified during SAQC, although this includes many legitimate extreme events that are crucial for calibrating and verifying the tail end of atmospheric behaviour that can have the largest societal impact. These results are generally on par with the percentage of keying errors identified in similar digitization efforts (Brönnimann et al., 2006).