# Peer review of "A rescued dataset of sub-daily meteorological observations for Europe and the southern Mediterranean region, 1877–2012"

_Earth System Science Data, 2018_

## Referee Comment (RC1) · Anonymous Referee #1 · 28 May 2018

**Review of "A rescued dataset of sub-daily meteorological observations for Europe and the southern Mediterranean region, 1877-2012" by Linden Ashcroft and colleagues**

This discussion paper outlines a collection of rescued data under the auspices of the FP7 UERRA project. The rescued data is of clear relevance and importance to a broad community of stakeholders. The data collection is described in a manner that is generally accessible to the interested expert (although see a number of comments and suggestions below). Given these considerations and that the subject matter is clearly within journal scope I would recommend acceptance of this paper following addressing a number of specific points and queries outlined below.

**Major points**

1. I'm not sure I would agree that snow depth and snowfall are non-ECVs (p.3). As noted in the GCOS status report (I believe GCOS-194 but am writing this review offline) the ECVs can each cover multiple variables. I believe these would be treated as sub-classes of the precipitation ECV in this context. Equally, there are one or more terrestrial ECVs under which these parameters could plausibly fall. I would suggest checking this with GCOS secretariat.

2. I would appreciate clarification on a number of methodological points to assure the ability of a reader to replicate / fully understand your chosen approach as follows:

   a. Does the "key as you see" approach extend to the keying of obviously incorrect entries or did the digitizers instead correct what they saw as unambiguous errors? If the latter what guidance was given? Regardless please be more explicit in the text p.5 lines 7-10. If it is strict key as you see this is hard to reconcile with the description of results given in Section 4.2. If I am confused so will your readers be.

   b. Assuming that the pages were in a variety of languages what supports were given to digitizers to account for this? This perhaps would be best addressed by an addition to the paragraph p.5 lines 19-22

   c. Assuming there is either a paper or a technical note describing the automated QA described in P.7 line 15 please cite it. Otherwise for strict repeatability you need to considerably expand this section (3.2) so that it adequately describes the exact chosen approaches or add a technical appendix covering this.

   d. Section 3.4 is a stub section and does not provide sufficient context to the reader. The reader is suddenly confronted with a flag value of 3 but in the prior text it was never mentioned what flag values were and why they were used.

   The above points perhaps collectively allude to the value of the authors, with what will now be fresh eyes, reviewing anew the methodological descriptions and ensuring that they fully reflect the methodological details of the work undertaken such that a non-participant could replicate their work. This revision should be in a manner that is accessible, understandable, and comprehensive. My feeling is that details glaringly obvious to the authors, who have lived and breathed this work for a number of years, but opaque to outsiders, have been omitted.

3. It is important to be clear whether your quality assessment approach is to flag and remove or flag and retain. No quality assurance program is perfect and removing values removes the ability to revisit QA choices in future. So, I hope and trust that it is flag and retain and not flag and remove, or that at a minimum the original digitized values are available somehow. Please address this point in a revised opening to Section 3. Either it is flag and retain or you must make the raw digitized data available and document how this can be accessed to assure the longest-term value of the collection effort being described.

4. Nowhere is it mentioned whether the original images are made available anywhere (except a brief and very non-specific reference in the caption to figure 2 in an 'on request' mode which has an implicit limited lifetime of applicability). Given the lack of duplicate keying in many cases, together with issues identified in several places, I would have thought that maintaining an archive of these images would be invaluable to researchers potentially many years hence using this collection. I would therefore urge making the images available via a sticky doi and ensuring this is documented in the final version of the manuscript.

5. Figure 1 dots are very unclear against a red background and probably illegible to color-blind folks. Please revise the figure so it is much clearer and use a color-blind simulator (of which several are available) to assure accessibility prior to resubmission.

6. Figure 10 introduces a whole bunch of flag codes which are utterly incomprehensible without context. Either change the key so its intuitive labels for each case or augment the figure caption to state what each flag code means in a human understandable manner (descriptive not codes).

7. I would urge giving geographically meaningful names for each region in figure 11 (and thus table 6). Using geographically meaningful names would enable figure 11 to standalone and thus increase its value.

**Minor points**
1. P.1 line 17 millions of such (add of)
2. P.2 line 18 and / or (add /)
3. P.2 line 21 economically rather than economic
4. P.6 line 14 replace variables with events – it is the events that are extremes not the variables!
5. P.6 line 28 still under development (under, not in)
6. P.8 line 19 high spatial separations (not distance)
7. P.8 line 23 large spatial separation
8. P.8 line 27 a subset of observing times … I think would make more sense?
9. P.9 line 37 Despite being amongst the countries with (given that you go on to describe 2 countries!)
10. P.10 line 10 I would remove the paragraph break here for readability personally.
11. P.10 line 36 tails of atmospheric behaviour …
12. P.13 line 18 data keyed by others …

13. P.13 line 29 remove the qualitative rider 'totally'
14. P.14 line 13 the C3S 311a Lot 2 Global Land and Marine Observations Database service contract through …
15. Peterson et al reference is missing a year in the reference list
16. Personal taste but I would present Figure 9 as something other than a pie-chart that can illicit some pretty strong reactions amongst the scientific community. There is almost always a better way to show such results in my (perhaps jaded) experience.

---

## Referee Comment (RC2) · Anonymous Referee #2 · 6 Jun 2018

**Review of "A rescued dataset of sub-daily meteorological observations for Europe and the southern Mediterranean region, 1877-2012" by Linden Ashcroft et al.**

The paper by Ashcroft et al. describes their strategy and approaches to rescue sub-daily meteorological observations within the UERRA project. They address the important issue of large amounts of undigitized data that are in danger of being lost completely. The resulting dataset contains a wide range of relevant variables. These data are of great value for the science community. Besides providing valuable data, the work contributes to foster data rescue and to improve data rescue approaches. Therefore, I recommend acceptance of the paper after several important corrections and clarifications.

**Major comments**

1. There seem to be some inconsistencies in the dataset. Below I show some examples of what I found:

   a. In some files, days without measurements are not shown:

```
ALG0060355  SkiddaCapBougarouni  60355  37.08  6.47  195  so04  28    28    28    28    3.0.0  fl30  nc  1931-06-29T07:00
ALG0060355  SkiddaCapBougarouni  60355  37.08  6.47  195  so04  22.4  22.4  22.4  22.4  3.0.0  fl30  nc  1931-06-30T07:00
ALG0060355  SkiddaCapBougarouni  60355  37.08  6.47  195  so04  35    35                3.0.0  fl36  nc  1931-07-02T07:00
ALG0060355  SkiddaCapBougarouni  60355  37.08  6.47  195  so04  26    26                3.0.0  fl36  nc  1931-07-03T07:00
ALG0060355  SkiddaCapBougarouni  60355  37.08  6.47  195  so04  25.6  25.6              3.0.0  fl36  nc  1931-07-04T07:00
ALG0060355  SkiddaCapBougarouni  60355  37.08  6.47  195  so04  25.4  25.4              3.0.0  fl36  nc  1931-07-05T07:00
ALG0060355  SkiddaCapBougarouni  60355  37.08  6.47  195  so04  25.6  25.6              3.0.0  fl36  nc  1931-07-11T07:00
ALG0060355  SkiddaCapBougarouni  60355  37.08  6.47  195  so04  32.4  32.4              3.0.0  fl36  nc  1931-07-15T07:00
ALG0060355  SkiddaCapBougarouni  60355  37.08  6.47  195  so04  23    23                3.0.0  fl36  nc  1931-07-16T07:00
```

   In some other cases, the time steps are continuous, but there is no raw data shown:

```
SLV0014249  NovoMesto  99999  45.8  15.18  220  so65  13  13  13  13  3.0.0  fl10  nc  1964-12-29T99:99
SLV0014249  NovoMesto  99999  45.8  15.18  220  so65   9   9   9   9  3.0.0  fl10  nc  1964-12-30T99:99
SLV0014249  NovoMesto  99999  45.8  15.18  220  so65   9   9   9   9  3.0.0  fl10  nc  1964-12-31T99:99
SLV0014249  NovoMesto  99999  45.8  15.18  220  so65                  3.0.0  fl13  nc  1965-01-01T99:99
SLV0014249  NovoMesto  99999  45.8  15.18  220  so65                  3.0.0  fl13  nc  1965-01-02T99:99
SLV0014249  NovoMesto  99999  45.8  15.18  220  so65                  3.0.0  fl13  nc  1965-01-03T99:99
SLV0014249  NovoMesto  99999  45.8  15.18  220  so65                  3.0.0  fl13  nc  1965-01-04T99:99
SLV0014249  NovoMesto  99999  45.8  15.18  220  so65                  3.0.0  fl13  nc  1965-01-05T99:99
SLV0014249  NovoMesto  99999  45.8  15.18  220  so65                  3.0.0  fl13  nc  1965-01-06T99:99
SLV0014249  NovoMesto  99999  45.8  15.18  220  so65                  3.0.0  fl13  nc  1965-01-07T99:99
SLV0014249  NovoMesto  99999  45.8  15.18  220  so65                  3.0.0  fl13  nc  1965-01-08T99:99
SLV0014249  NovoMesto  99999  45.8  15.18  220  so65                  3.0.0  fl13  nc  1965-01-09T99:99
SLV0014249  NovoMesto  99999  45.8  15.18  220  so65                  3.0.0  fl13  nc  1965-01-10T99:99
SLV0014249  NovoMesto  99999  45.8  15.18  220  so65                  3.0.0  fl13  nc  1965-01-11T99:99
SLV0014249  NovoMesto  99999  45.8  15.18  220  so65                  3.0.0  fl13  nc  1965-01-12T99:99
SLV0014249  NovoMesto  99999  45.8  15.18  220  so65                  3.0.0  fl13  nc  1965-01-13T99:99
SLV0014249  NovoMesto  99999  45.8  15.18  220  so65                  3.0.0  fl13  nc  1965-01-14T99:99
SLV0014249  NovoMesto  99999  45.8  15.18  220  so65  14              3.0.0  fl13  nc  1965-01-15T99:99
SLV0014249  NovoMesto  99999  45.8  15.18  220  so65                  3.0.0  fl13  nc  1965-01-16T99:99
SLV0014249  NovoMesto  99999  45.8  15.18  220  so65                  3.0.0  fl13  nc  1965-01-17T99:99
```

   In the second case, not showing raw data but using the flag fl13 is not compatible. I would suggest to always use continuous time steps within each time series and to include a code for "missing value" and for "test not applied". Continuous time steps and completely filled fields in the table will be much more user friendly.

   b. In some files, undefined numbers/texts appear in the column of the raw data:

```
ALG0060357  AnnabaCapdeGarde  60357  36.97  7.79  161  so04  29.8     29.8  29.8  29.8  3.0.0  fl30  nc  1922-08-07T07:00
ALG0060357  AnnabaCapdeGarde  60357  36.97  7.79  161  so04  NO DATA                     3.0.0  fl11  nc  1922-08-13T07:00
ALG0060357  AnnabaCapdeGarde  60357  36.97  7.79  161  so04  26.2     26.2  26.2  26.2  3.0.0  fl30  nc  1922-08-15T07:00
SLV0014249  NovoMesto  99999  45.8  15.18  220  so65  2      2     2     2     3.0.0  fl10  nc  1975-03-31T07:00
SLV0014249  NovoMesto  99999  45.8  15.18  220  so65  42-44                   3.0.0  fl17  nc  1975-08-01T07:00
SLV0014249  NovoMesto  99999  45.8  15.18  220  so65  56-58                   3.0.0  fl17  nc  1975-08-01T19:00
SLV0014249  NovoMesto  99999  45.8  15.18  220  so65  0      0     0     0     3.0.0  fl10  nc  1976-01-01T07:00
EGY0062342  ElMansura  62342  31.34  31.08  10  so16  23    23  23  23  3.0.0  fl30  nc  1953-05-30T18:00
EGY0062342  ElMansura  62342  31.34  31.08  10  so16  25    25  25  25  3.0.0  fl30  nc  1953-05-31T06:00
EGY0062342  ElMansura  62342  31.34  31.08  10  so16  -100              3.0.0  fl17  nc  1953-05-31T12:00
EGY0062342  ElMansura  62342  31.34  31.08  10  so16  -100              3.0.0  fl17  nc  1953-05-31T18:00
EGY0062342  ElMansura  62342  31.34  31.08  10  so16  -100              3.0.0  fl17  nc  1953-06-01T06:00
EGY0062342  ElMansura  62342  31.34  31.08  10  so16  -100              3.0.0  fl17  nc  1953-06-01T12:00
```

   c. For some files, the station ID seems to be corrupt:

```
ALG10XXXXX  FortNational  99999  36.63  4.2  942  so04  16    16    16    16    3.0.0  fl30  nc  1938-10-24T07:00
ALG10XXXXX  FortNational  99999  36.63  4.2  942  so04  16    16    16    16    3.0.0  fl30  nc  1938-10-25T07:00
ALG10XXXXX  FortNational  99999  36.63  4.2  942  so04  12.3  12.3  12.3  12.3  3.0.0  fl30  nc  1938-10-26T07:00
ALG10XXXXX  FortNational  99999  36.63  4.2  942  so04  10.2  10.2  10.2  10.2  3.0.0  fl30  nc  1938-10-27T07:00
```

   d. For wind direction, there appears a "C" and a "0" in some files:

```
CRO0014236    Zagreb-Gric    14236    45.82    15.98    157    so56    ESE    ESE    ESE    112.5    3.0.0    fl10    16ps    1949-01-27T14:00
CRO0014236    Zagreb-Gric    14236    45.82    15.98    157    so56    C      C      C      0        3.0.0    fl10    16ps    1949-01-27T21:00
CRO0014236    Zagreb-Gric    14236    45.82    15.98    157    so56    WNW    WNW    WNW    292.5    3.0.0    fl10    16ps    1949-01-28T07:00
CRO0014236    Zagreb-Gric    14236    45.82    15.98    157    so56    S      S      S      180      3.0.0    fl10    16ps    1949-01-28T14:00
CRO0014236    Zagreb-Gric    14236    45.82    15.98    157    so56    C      C      C      0        3.0.0    fl10    16ps    1949-01-28T21:00
CRO0014236    Zagreb-Gric    14236    45.82    15.98    157    so56    C      C      C      0        3.0.0    fl10    16ps    1949-01-29T07:00
CRO0014236    Zagreb-Gric    14236    45.82    15.98    157    so56    S      S      S      180      3.0.0    fl10    16ps    1949-01-29T14:00
CRO0014236    Zagreb-Gric    14236    45.82    15.98    157    so56    C      C      C      0        3.0.0    fl10    16ps    1949-01-29T21:00
CRO0014236    Zagreb-Gric    14236    45.82    15.98    157    so56    C      C      C      0        3.0.0    fl10    16ps    1949-01-30T07:00
CRO0014236    Zagreb-Gric    14236    45.82    15.98    157    so56    SW     SW     SW     225      3.0.0    fl10    16ps    1949-01-30T14:00
ALG0060355    SkiddaCapBougarouni    60355    37.08    6.47    195    so04    WSW    WSW    WSW    247.5    3.0.0    fl30    16ps    1933-03-30T07:00
ALG0060355    SkiddaCapBougarouni    60355    37.08    6.47    195    so04    0      0      0      0        3.0.0    fl30    16ps    1933-03-31T07:00
ALG0060355    SkiddaCapBougarouni    60355    37.08    6.47    195    so04    0      0      0      0        3.0.0    fl30    16ps    1933-04-01T07:00
ALG0060355    SkiddaCapBougarouni    60355    37.08    6.47    195    so04    E      E      E      90       3.0.0    fl30    16ps    1933-04-02T07:00
ALG0060355    SkiddaCapBougarouni    60355    37.08    6.47    195    so04    0      0      0      0        3.0.0    fl30    16ps    1933-04-03T07:00
ALG0060355    SkiddaCapBougarouni    60355    37.08    6.47    195    so04    0      0      0      0        3.0.0    fl30    16ps    1933-04-04T07:00
ALG0060355    SkiddaCapBougarouni    60355    37.08    6.47    195    so04    0      0      0      0        3.0.0    fl30    16ps    1933-04-05T07:00
ALG0060355    SkiddaCapBougarouni    60355    37.08    6.47    195    so04    W      W      W      270      3.0.0    fl30    16ps    1933-04-06T07:00
ALG0060355    SkiddaCapBougarouni    60355    37.08    6.47    195    so04    W      W      W      270      3.0.0    fl30    16ps    1933-04-07T07:00
ALG0060355    SkiddaCapBougarouni    60355    37.08    6.47    195    so04    W      W      W      270      3.0.0    fl30    16ps    1933-04-08T07:00
ALG0060355    SkiddaCapBougarouni    60355    37.08    6.47    195    so04    0      0      0      0        3.0.0    fl30    16ps    1933-04-09T07:00
ALG0060355    SkiddaCapBougarouni    60355    37.08    6.47    195    so04    0      0      0      0        3.0.0    fl30    16ps    1933-04-10T07:00
```

Are these codes for "no wind"? If yes, there should be a single code only, and it should not result in a "0" in the columns "convertedvalue" (0°=360°). I recommend to include a special code for "no wind".

e. Please clean the database from such inconsistencies. I recommend to create and run specific tests for each column in order to detect and correct cases as shown above.

2. For selecting areas and time periods to digitize observations the authors used the MARS. In the text, questions remain regarding these choices:

a. In the text, it does not get clear how comprehensive the data collected in the MARS are. Are there other digitized datasets that are not yet included in MARS (and could fill gaps more efficiently than digitizing analog data sources)? Did you search for such datasets?

b. Give some examples on how your work affects the MARS. How much did the data completeness increase in certain areas and/or time periods? How much longer near-complete time series are available thanks to your efforts? A simple figure could demonstrate the benefit of the new dataset.

c. Furthermore, giving a brief greater context of data availability the investigated regions (e.g. demonstrating that observations from these early time period are quite unique) would highlight the importance of the newly digitized dataset.

d. Data users are normally interested in long-term station records. Providing some instructions on how data user can generate the most comprehensive time series possible would be very useful (E.g. where to find further digitized data to combine the newly digitized data with? In the MARS or are there other data sources available?).

3. Some questions regarding HQC:

a. You applied the HQC only on the original time scale of the observations, is that correct? Aggregating the observation to daily data may strongly increase the number of neighboring stations usable for spatial consistency checks.

b. Did you include other stations resp. time series for HQC or did you just use the data you digitized? Using as many time series as possible will strongly increase the efficiency of the test. Please define these points clearly in the text and explain your decisions.

c. Similarly, you applied the SAQC on the time scale of observations only? Do you think you could have identified more errors if aggregating the data on larger time scales (particularly on daily time scale) and run adapted SAQC tests again?

d. Most of the tests in HQC are not spatial consistency test (Table 5) and seem to be repetitions or additions to the SAQC tests. Please define the HQC more clearly and explain better why you choose to use these tests.

**Minor comments**

P2 line 15: delete "in Germany".

P4 line 5: Please indicate earlier in the text how you define sub-daily (half-hourly to daily) and/or refer earlier to Table 7 (e.g. here in 2.2).

P5 line 8: "key as you see" approach: How where the digitizers supposed to proceed in ambiguous cases such as unclear handwriting? Was there a code to mark unrecognizable fields?

P5 line 12: Add the WMOs "GUIDELINES ON BEST PRACTICES FOR CLIMATE DATA RESCUE" as a reference.

P5 line 17: What variations are you talking about here? Can you give an example?

P5 line 26: Delete "e.g." before "Fig. 3".

P6 line 3: Could you make available these metadata? They could be important particularly for subsequent data homogenization.

P6 line 5: A complete QC procedure is supposed to detect non-systematic errors in the data too in my opinion. A simple solution would be to delete "non-systematic".

P6 line 18: You imply that automated QC approaches include spatial consistency tests which is not necessarily the case.

P6 line 31: Who are these climatologists? Were they part of UERRA or did you work together with the National Weather Services? Did you do this tests on the all digitized data or at random?

P6 line 33: "… common digitization errors…"

P8 line 3: Please show what "HQC" abbreviates.

P8 line 7-11: It does not seem very logical that low spatial resolution and observations taken at inconsistent times lead to a large number of positive flags (I would expect some thresholds for correlation and the number of neighboring stations to apply the test on a time series). Please explain.

P8 line 9-11: Rephrase the sentence.

P8 line 12-14: I would expect that reducing these thresholds would allow to apply the test on more time series, but rather produce more false positives. Was the problem of using 10 neighbors too low station correlation, which would mean that correlation is not included in the test?

P8 line 14-15: These test are not spatial consistency checks. According to the description of HQC, I would expect spatial consistency tests only.

P8 line 29-31: It remains unclear what tests the final check includes and what the flag of 3 is about. Please explain more clearly.

P8 line 34 – P9 line 2: This paragraph is mostly a repetition.

P9 line 39-40: Was this digitizer part of the UERRA project? Why did you set the data to missing and did not use the appropriate flag for the error?

P10 line 6-9: How can changing frequency of observation skew resulting analysis (aggregating the hourly data should be sufficient)? What correction did you apply?

P10 line 10-13: This paragraph is not clear to me. Are the 6-hourly observations from Slovenia too and/or from other countries?

P10 line 14-20: This paragraph is not very clear to me. The case of the effects of WW2 on the data seems very interesting, but it should be explained a bit more how the specific problems may have occurred.

P10 line 21: Delete duplication of "Quality control result".

P10 line 23: I think the source resolution and temporal gaps should not lead to more frequent detections by the QC method?

P10 line 25: For clarity, I would call flags always "fl.." and not just by the number (here "36").

P10 line 28: Same as previous comment.

P10 line 31 and 34: What is described cannot be seen clearly in Fig. 12. See also my suggestions for adaptations of Fig. 12.

P10 line 32-33: Do you want to say that about the same amount of observations was flagged by spatial consistency tests like in other analyses?

P10 line 37-38: It is unclear to what the last sentence refers to.

P11 line 4-5: Meaning of prefix "4" remains rather unclear. Please define more clearly.

P11 line 36: "Triple, or even five-time keying…": Where do this numbers come from? Reference?

P12 line 4-5: Please reformulate the sentence to make it more clear.

P12 line 10-12: Even though the statement is probably generally true, can you conclude that (in such a strong for) from your work? The fraction of errors detected in your dataset is rather small.

P12 line 25-26: I think the recommendation of five-time keying applies rather to specific examples (such as these observation take at sea) than it can be considered as a general rule?

P12 line 29: Delete "and useful".

P12 line 34-39: What are the concrete conclusions of these assessments? Does it result in integration or removal of data sources? Correction/prevention of errors e.g. by re-scanning original data sources? Giving more specific instructions of digitizers?

P13 line 1-13: This paragraph is rather unclear and I suggest to restructure it. Suggestion: 1. Our findings in the case of Zagazig do not confirm WMOs recommendation regarding templates. Mention some advantages of using templates too. 2. The case highlights the importance of creating user friendly templates. 3. Recommendation of creating user friendly templates (e.g. include feedback of digitizers when designing templates).

P13 line 22-23: Delete "It is no use… already complete.".

P13 line 23-25: Suggestion for this sentence: "Conducting QC as soon as data become available means that the digitizers may detect own keying errors, and that you can advise them on how to increase the quality of their work.".

P13 line 26-31: Did you check for and detect such systematic quality issues? If yes, how was that included in your study respectively why not? If you did not check: Do you think such systematic quality issues are a potential problem in your dataset?

P14 line 6: "Manual checking of values and decisions based on expert knowledge may mean…"

P14 line 11: Replace "The digitized dataset" by "It".

Table 2: I suggest to always use the same number of decimals within one field.

Table 3: Do not use the same illustration for different tests (Climatic outliers, Big jumps and sharp spikes).

Table 3: I suggest to mention the change in the Duplicate values test only in the text and not in the table.

Table 3: Precipitation totals seems not to be a test but just a daily aggregation. Please specify or remove it from the table.

Table 4: Define more clearly to which test the flags relate to (e.g. does lf10 mean "Passed SAQC"?).

Table 5: I think this table could be moved to the supplementary material. Furthermore, only a few tests are spatial consistency test (HQC is described as QC for spatial consistency in the text).

Table 6: The caption of the table is unclear, please reformulate.

Table 6: Observing times are mostly not in accordance with observing times from Table 7. Please explain these differences.

Figure 1: Remove small titles above figures.

Figure 1: Years shown in small titles above figures (1960-2010) not in accordance with time period in legend (1950-2010).

Figure 2: Delete last sentence (same to find in the text) or correct it and provide a direct link to the images.

Figure 6: Is the title needed?

Figure 6: I also suggest to replace "Percentage of tested data %" by "Flagged observations [%]".

Figure 6: Show variable acronyms horizontally.

Figure 7: Correct last sentence in caption.

Figure 7: I suggest to remove longitude and latitude here.

Figure 7: An indication (or additional plot) of the length of the near-complete time series of these stations after filling data gaps within your study would be an interesting asset for data users.

Figure 8: remove "1e6" from plots.

Figure 8 d): Invert scale of hours of the day, separate "daily" more clearly from the rest would be helpful.

Figure 8: I suggest to replace "Data count" by "Observations".

Figure 9: I would remove the first pie-chart from the figure, describing the numbers in the text is sufficient.

Figure 9: It is not clear how you chose the flags shown here (% of each flag of all flagged values? Other flags too small to be represented? Very similar to Fig. 12?). Please clarify or remove the figure.

Figure 10 d): I do not think this figure provides much information, I suggest to remove it.

Figure 10: Remove "1e1" from figures.

Figure 10: I suggest to replace "Percentage of total data %" by "Flagged observations [%]".

Figure 10: Colors are hard to distinguish.

Figure 11: Remove title.

Figure 11: Replace "Percentage of tested data %" by "Flagged observations [%]".

Figure 12: Changing % between "Total" and "Flags" is unclear. I suggest to show the fraction of flagged observation of the total in ‰.

Figure 12: The order in "Flags" seems a bit random.

Figure 12: The color code is unclear. There are three darkness classes for each color, but in the caption only two classes are explained.

Figure 12: What group is "Passed SAQC but recheck required"? Why would you recheck observations that passed the SAQC test?

---

## Short Comment (SC1) · 7 Jun 2018

I was the developer of UQC (Universal Quality Control for hourly observations of surface air temperature, air humidity, wind speed and wind direction), a unit of the SAQC software package. Reviewer 2 asks (at point 3c) if aggregating the hourly data to daily would help to identify more errors, and one could run adapted SAQC with aggregated data on daily scale. The concept of developing SAQC was to identify transcription errors, measurement unit errors and systematic errors of relatively large magnitudes. Transcription errors (which are the most frequent) are best appear on hourly scale, as aggregating 1 wrong value with one or more several correct values will result in a sum

whose deviation from the expected value is much smaller than that of the wrong hourly value. On the other hand, SAQC includes examinations for the detection of systematic errors, they are the control of mean absolute increment between successive data and the control of the frequency distribution of the data (they are explained in Table 3 of the study, together with the other examinations). The Manual (UQC_manual.pdf), which includes both guide and scientific description and accessible via the webpage indicated in the study (page 7, line 15, it is an answer also to Reviewer 1's, point 2c), does not offer the option to use the software with daily data. However, with the assignation of any arbitrary hour to daily data, the program could be applied to daily data. We did not do this, as we focused on the most frequently occurring errors in a dataset of more than 8 million hourly records. I agree with Reviewer 2 that likely more errors could be identified with widening the scale of quality control examinations, but with respect to the size of the dataset we had to stop the controls at a certain point. I hope that in spite of some unexploited options of possible further QC examinations, the QC procedures accomplished and especially SAQC is a positive contribution to the dataset development described in our study.

---

## Author Comment (AC1) · 25 Jul 2018

We thank the reviewers for their considered and detailed responses. We have endeavoured to reply to all points below, and have incorporated the majority of the suggestions made in the revised manuscript.

**Anonymous referee #1 comments**

| |
|---|
| **Review of "A rescued dataset of sub-daily meteorological observations for Europe and the southern Mediterranean region, 1877-2012" by Linden Ashcroft and colleagues** |
| *This discussion paper outlines a collection of rescued data under the auspices of the FP7 UERRA project. The rescued data is of clear relevance and importance to a broad community of stakeholders. The data collection is described in a manner that is generally accessible to the interested expert (although see a number of comments and suggestions below). Given these considerations and that the subject matter is clearly within journal scope I would recommend acceptance of this paper following addressing a number of specific points and queries outlined below.* |
| Many thanks to the reviewer for such a thorough examination of our manuscript. We are very grateful for their comments! |
| **Major points** |
| **1.** *I'm not sure I would agree that snow depth and snowfall are non-ECVs (p.3). As noted in the GCOS status report (I believe GCOS-194 but am writing this review offline) the ECVs can each cover multiple variables. I believe these would be treated as sub-classes of the precipitation ECV in this context. Equally, there are one or more terrestrial ECVs under which these parameters could plausibly fall. I would suggest checking this with GCOS secretariat.* |
| Thanks for picking this up, we have updated the manuscript accordingly. In particular, lines 20–24 of page 3 have been updated to read:

"The variables of interest were several atmospheric and terrestrial Essential Climate Variables (ECVs) as defined by the Global Climate Observing System (GCOS, World Meteorological Organization, 2015): air temperature (TT), atmospheric pressure (sea level pressure, PP and station level pressure, SP), wind speed (WS) and wind direction (WD), relative humidity (RH), dew point temperature (DP) and daily rainfall (RR), fresh snowfall (FS), and snow depth (SD)." |
| **2.** *I would appreciate clarification on a number of methodological points to assure the ability of a reader to replicate / fully understand your chosen approach as follows:*
**a.** *Does the "key as you see" approach extend to the keying of obviously incorrect entries or did the digitizers instead correct what they saw as unambiguous errors? If the latter what guidance was given? Regardless please be more explicit in the text p.5 lines 7-10. If it is strict key as you see this is hard to reconcile with the description of results given in Section 4.2. If I am confused so will your readers be.* |
| This text has been updated to more accurately reflect the methodology: " Clear, unambiguous, errors in the data sources were generally retained by the digitisers and recorded in station metadata files which were later used when quality controlling the data (see Sect. 3). If a digitizer could not read a value due to poor handwriting or |

scanning issues, they represented it by a value of -88.8, while missing values were set to -99.9."

***b.*** *Assuming that the pages were in a variety of languages what supports were given to digitizers to account for this? This perhaps would be best addressed by an addition to the paragraph p.5 lines 19-22*

We have added the following information to the suggested section: "Translation of the relevant column and row headings was provided to the digitizers for each source, as well as the various wind strength scales."

***c.*** *Assuming there is either a paper or a technical note describing the automated QA described in P.7 line 15 please cite it. Otherwise for strict repeatability you need to considerably expand this section (3.2) so that it adequately describes the exact chosen approaches or add a technical appendix covering this.*

The relevant text has been updated to provide more information on the SAQC procedure:

"Statistical quality control was conducted using a semi-automatic quality control (SAQC) procedure (Universitat Rovira i Virgili, 2014). The SAQC method was largely adapted from existing automatic quality control procedures developed for sub-daily data at a global scale (e.g. Dunn et al., 2012; Durre et al., 2010), but was modified for the UERRA dataset to enable more manual examination of the resultant flags. Full details of the procedure, the relevant software and instructions for use are available from "A.Q.C. Software" menu from http://www.c3.urv.cat/softdata.php."

***d.*** *Section 3.4 is a stub section and does not provide sufficient context to the reader. The reader is suddenly confronted with a flag value of 3 but in the prior text it was never mentioned what flag values were and why they were used.*

Many thanks for pointing this out, we have removed section 3.4 and incorporated this information in section 4.4 which details the additional quality control checking: "In the final data check, a small conversion problem was detected with the sea level pressure at two Slovenian stations (around 318,000 values). The vast majority of these observations passed both SAQC and HQC, with large errors identified and flagged appropriately. However, these observations were marked with a prefix of "4" rather than "1" (subjected to SAQC) or "3" (subjected to SAQC and HQC) in the final dataset, to signify that additional QC may be required by future users."

*The above points perhaps collectively allude to the value of the authors, with what will now be fresh eyes, reviewing anew the methodological descriptions and ensuring that they fully reflect the methodological details of the work undertaken such that a non-participant could replicate their work. This revision should be in a manner that is accessible, understandable, and comprehensive. My feeling is that details glaringly obvious to the authors, who have lived and breathed this work for a number of years, but opaque to outsiders, have been omitted.*

Many thanks for this insightful point. We have reread the manuscript with clear eyes and made a number of clarifications and adjustments that we hope make the methodology clearer.

*3. It is important to be clear whether your quality assessment approach is to flag and remove or flag and retain. No quality assurance program is perfect and removing values removes the ability to revisit QA choices in future. So, I hope and trust that it is flag and retain and not flag and remove, or that at a minimum the original digitized values are available somehow. Please address this point in a revised opening to Section 3. Either it is flag and retain or you must make the raw digitized data available and document how this can be accessed to assure the longest-term value of the collection effort being described.*

The original data are retained, and in fact we provide four versions of the dataset in PANGAEA: the original data in its original units (V1), the data after SAQC has been applied (V2), the data after SAQC and HQC have been applied (V3), and the values from V3 converted to SI units (convalue). While this is mentioned in section 6 we agree that is has not been made clear in section 3, and so have updated both sections accordingly.

*4. Nowhere is it mentioned whether the original images are made available anywhere (except a brief and very non-specific reference in the caption to figure 2 in an 'on request' mode which has an implicit limited lifetime of applicability). Given the lack of duplicate keying in many cases, together with issues identified in several places, I would have thought that maintaining an archive of these images would be invaluable to researchers potentially many years hence using this collection. I would therefore urge making the images available via a sticky doi and ensuring this is documented in the final version of the manuscript.*

In section 6 (data availability) we note that " The original data scans are available through each data repository (Table S2) and through the Universitat Rovira i Virgili Centre for Climate Change (ftp://130.206.36.123, user: C3_UERRA, password: c3uerra17)." We have also updated the caption of Figure 2 to reflect this.

*5. Figure 1 dots are very unclear against a red background and probably illegible to color-blind folks. Please revise the figure so it is much clearer and use a color-blind simulator (of which several are available) to assure accessibility prior to resubmission.*

We have updated this figure to be more colour-blind friendly, many thanks for the suggestion.

[Figure]

Updated Figure 1, showing the distribution of sub-daily observation in MARS.

**6.** *Figure 10 introduces a whole bunch of flag codes which are utterly incomprehensible without context. Either change the key so its intuitive labels for each case or augment the figure caption to state what each flag code means in a human understandable manner (descriptive not codes).*

We have updated the legend of Figure 10 to make it easier to understand out of context.

**7.** *I would urge giving geographically meaningful names for each region in figure 11 (and thus table 6). Using geographically meaningful names would enable figure 11 to standalone and thus increase its value.*

Great idea! We have updated the network names.

**Minor points**

*1. P.1 line 17 millions of such (add of)*
*2. P.2 line 18 and / or (add /)*
*3. P.2 line 21 economically rather than economic*
*4. P.6 line 14 replace variables with events – it is the events that are extremes not the variables!*
*5. P.6 line 28 still under development (under, not in)*
*6. P.8 line 19 high spatial separations (not distance)*
*7. P.8 line 23 large spatial separation*
*8. P.8 line 27 a subset of observing times … I think would make more sense?*
*9. P.9 line 37 Despite being amongst the countries with (given that you go on to describe 2 countries!)*

*10. P.10 line 10 I would remove the paragraph break here for readability personally.*
*11. P.10 line 36 tails of atmospheric behaviour …*
*12. P.13 line 18 data keyed by others …*
*13. P.13 line 29 remove the qualitative rider 'totally'*

We have corrected these typographical errors and thank the reviewer for reading the manuscript so carefully.

*14. P.14 line 13 the C3S 311a Lot 2 Global Land and Marine Observations Database service contract through …*

Updated.

*15. Peterson et al reference is missing a year in the reference list*

Updated.

*16. Personal taste but I would present Figure 9 as something other than a pie-chart that can illicit some pretty strong reactions amongst the scientific community. There is almost always a better way to show such results in my (perhaps jaded) experience.*

Based on this comment and those of reviewer 2 we have modified Figure 9 to be a bar chart, with more emphasis on the distribution of the error flags rather than the relative number of error flags in relation to the full dataset.

[Figure]

Updated Figure 9, showing the distribution of quality flags after the application of semi-automatic quality control (SAQC).

**Anonymous Referee #2 comments**

*The paper by Ashcroft et al. describes their strategy and approaches to rescue sub-daily meteorological observations within the UERRA project. They address the important issue of large amounts of undigitized data that are in danger of being lost completely. The resulting dataset contains a wide range of relevant variables. These data are of great value for the science community. Besides providing valuable data, the work contributes to foster data rescue and to improve data rescue approaches. Therefore, I recommend acceptance of the paper after several important corrections and clarifications.*

We thank the reviewer for their detailed review of the paper and the dataset.

**Major comments**

*1. There seem to be some inconsistencies in the dataset. Below I show some examples of what I found:*
*a. In some files, days without measurements are not shown: In some other cases, the time steps are continuous, but there is no raw data shown: In the second case, not showing raw data but using the flag fl13 is not compatible. I would suggest to always use continuous time steps within each time series and to include a code for "missing value" and for "test not applied". Continuous time steps and completely filled fields in the table will be much more user friendly.*

Many thanks for identifying this issue. We have reviewed the dataset and removed all time steps with no data. This in effect removes the issue you mentioned about the flag fl13 being 'applied' to no data. We also identified several values with the flag fl36 and no raw data values.

While we agree that in some cases having continuous time steps would be more useful for some data users, infilling all datasets with missing values to ensure continuity would require a large amount of extra space, not to mention a lot of extra work for the data formatting team at PANGAEA. To be consistent, we have instead opted to remove the 19093 time steps with no data, and trust that future users of the data can map the observations to the temporal resolution they require.

*b. In some files, undefined numbers/texts appear in the column of the raw data:*

We agree that there are some strange characters and values in the raw data (version 1 of the dataset), including the occasional errant comment from the digitisers. However, removing these strange values would reduce the transparency of the digitisation process, and prevent future users from building on our efforts. On balance, we decided to retain the undefined values in the raw version of the dataset, and have updated the data description (section 6) to include the following: "Version 1 contains the raw digitised data, including all typographical errors and other issues subsequently identified in the quality control procedure. We have retained this information to ensure transparency of the process, in case it is useful for future users of the dataset."

*c. For some files, the station ID seems to be corrupt:*

The stationIDs that include Xs represent stations that do not have a WMO number. We implemented this method to avoid confusion with the WMO station number coding system, and there are nine stations with this issue:

- ALG10XXXXX, Fort National

- CZE00XXXXX, Praded

- CZE10XXXXX, SlaknatePleso

- ESP10XXX42, Tarragona

- ESPXX00201, Barcelona

- ESPXX0200D, Barcelona Turo Del Putxet

- ESPXX0201B, Barcelona Atarazanas

- ESPXX9771C, Lleida

- TUR10XXX00, Karakose

Given that the station IDs already include characters (rather than being purely numeric), we don't think that this will affect any future analysis of the dataset and so have retained these unique IDs. WMO numbers are provided in a separate column, and stations with no WMO numbers are represented in this column by 99999.

*d. For wind direction, there appears a "C" and a "0" in some files: Are these codes for "no wind"? If yes, there should be a single code only, and it should not result in a "0" in the columns "convertedvalue" (0°=360°). I recommend to include a special code for "no wind".*

Yes, these are codes for 'no wind'. This discrepancy comes from the range of different sources used and we agree that it can lead to some confusion. However, it is clear from the metadata that 0 in the convalue column represents calm in the wind speed dataset. On advice from the PANGAEA data technicians, we have also retained the original combination of 0 and C in versions 1 to 3 of the dataset, as updating this error would take several months.

*e. Please clean the database from such inconsistencies. I recommend to create and run specific tests for each column in order to detect and correct cases as shown above.*

Many thanks again for taking such a close look at the dataset. We have implemented these suggestions, and have removed 19093 lines with no data in any version of the dataset (i.e. no data digitised by the original source). These data have been provided to PANGAEA, but may take several weeks to be updated on their public repository.

*2. For selecting areas and time periods to digitize observations the authors used the MARS. In the text, questions remain regarding these choices:*
*a. In the text, it does not get clear how comprehensive the data collected in the MARS are. Are there other digitized datasets that are not yet included in MARS (and could fill gaps more efficiently than digitizing analog data sources)? Did you search for such datasets?*

It is important to note here that the primary goal of this data rescue activity was to recover data that could be used in the improvement of European reanalyses, rather

than the recovery of long-term instrumental records. MARS is the primary data source for the majority of European reanalysis products (e.g. Dahlgren et al., 2016).

In section 2 we describe the method of identifying data that had not yet been digitised which involved the MARS archive, ISPD, ECA&D and the holdings of several countries in central Europe and the western Balkans regions. We also used the knowledge from the author team who have been involved in a number of data rescue activities for the European sector.

ISPD and ECA&D were explored first, as their online data exploration tools and catalogues are more publicly available. From these portals we could gain a visual understanding of data availability. Interrogating MARS is not as simple, so once we had visually identified regions lacking in sub-daily data (Scandinavia, Eastern Europe and the Mediterranean) we then extracted information from MARS about these regions. We selected the regions of interest based on the percentage of sub-daily data available for each station.

We also note that MARS annually receives station data from ISPD, and generally does not accept surface observations from any other databank. This means that while there may be other rescued datasets that are currently not in MARS, they need to be in ISPD first. This process, and the potential for duplicated effort, will hopefully be improved with the development of the Copernicus Climate Change Services Data Rescue Service and Climate Data Store (C3S DRS: http://about-c3s-dr.eu/).

We have updated section 2.1 to clarify this point, correcting some finer details of the searching process we used and explicitly stating the difficulties involved in interrogating MARS.

*b. Give some examples on how your work affects the MARS. How much did the data completeness increase in certain areas and/or time periods? How much longer near complete time series are available thanks to your efforts? A simple figure could demonstrate the benefit of the new dataset.*

The aim of this data rescue effort was to provide increased sub-daily coverage for European regional reanalyses, rather than developing long, complete time series. We focussed on infilling spatial gaps rather than temporal gaps, and the benefit of this data rescue effort will best be shown in the improvement of future regional reanalyses. We could replot Figure 1 of the paper and update the colours of some of the dots on these maps, but we believe that Figure 7 shows more useful information about the variables, observing times and length of the records.  We have clarified this in section 2.1 by adding the following: " Our aim was not to develop single, long-term data series for particular stations, but rather improve the availability of sub-daily observations anywhere that may be underrepresented in the current observational data used for European reanalysis products."

*c. Furthermore, giving a brief greater context of data availability the investigated regions (e.g. demonstrating that observations from these early time period are quite unique) would highlight the importance of the newly digitized dataset.*

The focus of this study is on filling spatial gaps in availability of sub-daily data in the post-1957 period to support the development of regional reanalyses. We can see that this is not very clear in the manuscript, and so have revised section 2 accordingly.

*d. Data users are normally interested in long-term station records. Providing some instructions on how data user can generate the most comprehensive time series possible would be very useful (E.g. where to find further digitized data to combine the newly digitized data with? In the MARS or are there other data sources available?).*

While we agree that long-term station records are very valuable for studies of climate variability, the data users for this project were more interested in high temporal resolution data networks rather than observations from a single point. Developing composite datasets is beyond the scope of the current work. Instead, post processing of the data in the development of reanalysis products will be able to merge observations and provide consistent long-term picture of climate variability.

To address this comment, we have added the following comment in section 6: " Through these repositories, future users should be able to develop long-term composite time series of these and other observations from the European sector."

*3. Some questions regarding HQC:*
*a. You applied the HQC only on the original time scale of the observations, is that correct? Aggregating the observation to daily data may strongly increase the number of neighboring stations usable for spatial consistency checks.*

Yes, that is correct. Analysing daily averages of the data would likely have increased the number of available neighbouring stations, but would have made it difficult to identify the individual erroneous values without manually checking the data at the sub-daily scale which would have been very time consuming. Additionally, the HadISD quality control procedure has been developed for sub-daily observations rather than daily averages. We have clarified this in section 3.3: "Only data that had been checked by visual and SAQC were subjected to this procedure and as with SAQC, the data were examined in their original temporal format to avoid removing valid data."

*b. Did you include other stations resp. time series for HQC or did you just use the data you digitized? Using as many time series as possible will strongly increase the efficiency of the test. Please define these points clearly in the text and explain your decisions.*

We have clarified this in section 3.3: "Due to time constraints, only data digitized as part of this project were used in the spatial quality assessment, although future work could make use of the existing HadISD dataset as a reference network."

*c. Similarly, you applied the SAQC on the time scale of observations only? Do you think you could have identified more errors if aggregating the data on larger time scales (particularly on daily time scale) and run adapted SAQC tests again?*

As discussed in an earlier interactive comment on this manuscript (https://doi.org/10.5194/essd-2018-39-SC1) from one of our authorship team, transcription errors (which are the most common error type) are best identified at the sub-daily scale. Aggregating one incorrect value with one or more correct values will

result in a sum whose deviation from the expected value is much smaller than that of the wrong hourly value. SAQC also includes examinations for the detection of systematic errors through the examination of frequency distributions of the data and mean absolute increment between successive data (as explained in Table 3). While we agree that examining the data on the daily time scale and rerunning SAQC may have identified additional errors, the size of the dataset (8 million observations) made this operationally difficult.

*d. Most of the tests in HQC are not spatial consistency test (Table 5) and seem to be repetitions or additions to the SAQC tests. Please define the HQC more clearly and explain better why you choose to use these tests.*

We take your point here and are grateful for the suggestion. HQC was conducted in addition to SAQC to provide some spatial testing of the data, but also act as a second round test for some of the most common errors that might have been missed by the SAQC. We offer users both versions of the dataset and they can choose which level of QC they would like to use. As we discuss later, the majority of the flags identified by HQC were related to neighbour checks.

We have amended this section to provide more information on HQC. Specifically, we now explicitly state that "The final QC procedure consisted of subjecting data from neighboring stations to spatial quality control tests, as well as rerunning several checks from SAQC in a fully automated way as a second-round check for gross errors that may have slipped through SAQC."

**Minor comments**

*P2 line 15: delete "in Germany".*

Done

*P4 line 5: Please indicate earlier in the text how you define sub-daily (half-hourly to daily) and/or refer earlier to Table 7 (e.g. here in 2.2).*

We have added the following definition to the introduction: "We define sub-daily variables here as variables observed at least once a day, up to every half an hour."

*P5 line 8: "key as you see" approach: How where the digitizers supposed to proceed in ambiguous cases such as unclear handwriting? Was there a code to mark unrecognizable fields?*

Based on this and comments from reviewer 1 we have updated this section to read: "Clear, unambiguous, errors in the data sources were generally retained by the digitisers and recorded in station metadata files which were later used when quality controlling the data (see Sect. 3). If a digitizer could not read a value due to poor handwriting or scanning issues, they represented it by a value of -88.8, while missing values were set to -99.9. "

*P5 line 12: Add the WMOs "GUIDELINES ON BEST PRACTICES FOR CLIMATE DATA RESCUE" as a reference.*

Done

*P5 line 17: What variations are you talking about here? Can you give an example?*

Updated to read: "Days with missing data were recorded in metadata files, along with any other variations in the data source, such as repeated pages in the scanned file or temporary changes in the file structure."

*P5 line 26: Delete "e.g." before "Fig. 3".*

Figure 3 shows some examples of the data sources and templates. We have updated this to read "see Fig. 3 for several examples".

*P6 line 3: Could you make available these metadata? They could be important particularly for subsequent data homogenization.*

Unfortunately only a small amount of metadata could be found for the sources, but this information is preserved in the final dataset, where changes to latitude, longitude and altitude are provided with each value. We have added the following sentence for clarification: "In most cases, only the station height information could be located, but any changes identified were recorded in the coordinates accompanying the final dataset (also available in Supplementary Table 2)."

*P6 line 5: A complete QC procedure is supposed to detect non-systematic errors in the data too in my opinion. A simple solution would be to delete "non-systematic".*

We assume you mean that a complete QC procedure is supposed to detect systematic errors as well. We have updated this section to read "Quality control (QC) procedures are crucial to identify non-systematic errors or shed light on systematic biases in the time series."

*P6 line 18: You imply that automated QC approaches include spatial consistency tests which is not necessarily the case.*

We've updated this sentence to read " However, a completely automated procedure that tests for spatial relationships, such as that used for global databases (Dunn et al., 2012) would also be sub-optimal, as the digitized data do not cover a wide geographic area and consistent time period."

*P6 line 31: Who are these climatologists? Were they part of UERRA or did you work together with the National Weather Services? Did you do this tests on the all digitized data or at random?*

The climatologists were postgraduate researchers from Centre for Climate Change at Universitat Rovira i Virgili, and the tests were done on data from all sources. We have updated section 3.1 to clarify this.

*P6 line 33: "… common digitization errors…"*

Corrected

*P8 line 3: Please show what "HQC" abbreviates.*

We've updated section 3.3 to clarify that HQC stands for Hadley quality control.

| |
|---|
| *P8 line 7-11: It does not seem very logical that low spatial resolution and observations taken at inconsistent times lead to a large number of positive flags (I would expect some thresholds for correlation and the number of neighboring stations to apply the test on a time series). Please explain.*
*P8 line 9-11: Rephrase the sentence.* |

When there are not enough neighbors or too many missing values in a series, the HadISD tests automatically removed all values, leading to a large number of positive flags.

We have revised this sentence to clarify: "Automatically running HQC in its standard form with the standard thresholds used in the development of the global HadISD dataset led to a large number of false positive flags being identified, automatically removing a significant number of correct observations from the dataset (Fig. 6)."

| |
|---|
| *P8 line 12-14: I would expect that reducing these thresholds would allow to apply the test on more time series, but rather produce more false positives. Was the problem of using 10 neighbors too low station correlation, which would mean that correlation is not included in the test?* |

Using ten neighboring stations required rather than five meant that more values were flagged as significantly different to their neighbors, despite all values being correct. We have updated this sentence to read: "To reduce the number of false positive flags and increase the number of stations that could be checked, the minimum number of neighboring stations required for HQC testing was reduced from ten to five, and the percentage of non-missing observations per month allowed was reduced from 75 % to 66 %."

| |
|---|
| *P8 line 14-15: These test are not spatial consistency checks. According to the description of HQC, I would expect spatial consistency tests only.* |

This is a valid point, and we have clarified the role of HQC in response to comment 3d. The HQC application did run a number of tests that were not looking at spatial consistency, to act as a second round of fully automated checking in addition to SAQC. The majority of the HQC flags were due to the neighbor checks, with some others coming from the wind and climatology tests. We suspect that the latter tests flagged values because the majority of the wind observations were at a coarse resolution (being converted from a Beaufort or similar scale) and a number of the stations were mountainous with the potential for climatologically extreme values.

| |
|---|
| *P8 line 29-31: It remains unclear what tests the final check includes and what the flag of 3 is about. Please explain more clearly.* |

This section has been removed and the information included in section 4.4 which describes the additional digitization quality assurance checks: "In the final data check, a small conversion problem was detected with the sea level pressure at two Slovenian stations (around 318,000 values).  The vast majority of these observations passed both SAQC and HQC, with large errors identified and flagged appropriately. However, these observations were marked with a prefix of "4" rather than "1" (subjected to SAQC) or

"3" (subjected to SAQC and HQC) in the final dataset, to signify that additional QC may be required by future users."

*P8 line 34 – P9 line 2: This paragraph is mostly a repetition.*

We have removed this sentence: " A small network of sources from Germany, Lebanon and Slovenia contained hourly data for a number of variables, contributing to the high number of observations for those countries."

*P9 line 39-40: Was this digitizer part of the UERRA project? Why did you set the data to missing and did not use the appropriate flag for the error?*

Many thanks for identifying this oversight. We did give the data a flag of fl11, and have updated the text accordingly.

*P10 line 6-9: How can changing frequency of observation skew resulting analysis (aggregating the hourly data should be sufficient)? What correction did you apply?*

We have updated this section to clarify: "This issue could significantly affect any future analysis of rainfall frequency using these data, and so these values were corrected, resulting in a number of fl12 (corrected based on original source) flags for rainfall."

*P10 line 10-13: This paragraph is not clear to me. Are the 6-hourly observations from Slovenia too and/or from other countries?*

Apologies for the confusion, the six-hourly observations are from the same Slovenian stations. This has been clarified in the text.

*P10 line 14-20: This paragraph is not very clear to me. The case of the effects of WW2 on the data seems very interesting, but it should be explained a bit more how the specific problems may have occurred.*

We have clarified this: "The higher number of fl17 flags (observations set to missing as no value could be found in the source image) during the 1940s may reflect data issues during the Second World War, particularly for Egypt and Algeria, where some original source files were ordered incorrectly. This resulted in a number of values being ascribed to the wrong date."

*P10 line 21: Delete duplication of "Quality control result".*

Thank you!

*P10 line 23: I think the source resolution and temporal gaps should not lead to more frequent detections by the QC method?*

This is a result of the HadISD/HQC method, which was designed to fully automate the quality control of a global dataset. If a station is missing a certain number of observations for a month, or too many stations in a network are missing data, then the whole period is flagged and removed. We have clarified this: "Temperature was the variable with the smallest number of flagged values overall by HQC, with the exception of network 2 (north Africa) where data source resolution and the high number of

missing values caused HQC to flag and remove extra values (Fig. 11). The variable with the highest proportion of flagged values in network 2 was sea level pressure."

*P10 line 25: For clarity, I would call flags always "fl.." and not just by the number (here "36").*

Updated

*P10 line 28: Same as previous comment.*

Updated

*P10 line 31 and 34: What is described cannot be seen clearly in Fig. 12. See also my suggestions for adaptations of Fig. 12.*

We've updated the reference to Fig. 12 here to clarify that we mean the dark orange section, and have also incorporated the later suggestions to improve this figure.

*P10 line 32-33: Do you want to say that about the same amount of observations was flagged by spatial consistency tests like in other analyses?*

Yes, we have clarified this.

*P10 line 37-38: It is unclear to what the last sentence refers to.*

We have clarified this: " These percentages of flagged values are similar to that identified by Brönnimann et al. (2006), who found key error rates of 0.2 % to 3 % for hourly temperature and upper air observations."

*P11 line 4-5: Meaning of prefix "4" remains rather unclear. Please define more clearly.*

We have slightly updated this section, but are not sure how to explain this additional prefix more clearly: "In the final data check, a small conversion problem was detected with the sea level pressure at two Slovenian stations (around 318,000 values). The vast majority of these observations passed both SAQC and HQC, with large errors identified and flagged appropriately. However, these observations were marked with a prefix of "4" rather than "1" (subjected to SAQC) or "3" (subjected to SAQC and HQC), to signify that additional QC may be required by future users."

*P11 line 36: "Triple, or even five-time keying…": Where do this numbers come from? Reference?*

These values came from the digitization methods employed by citizen science projects Old Weather and Weather Detective respectively, communicated directly by the project organisers. We agree that the values should be supported by peer-reviewed references though. In the absence of these, we have updated the statement to read: "Double entry, which is considered standard practise for many data entry activities (Barchard and Pace, 2011), would be the best way to overcome these issues, or even triple entry, which is the method used by a number of citizen science activities (e.g. Old Weather, www.oldweather.org)."

*P12 line 4-5: Please reformulate the sentence to make it more clear.*

This sentence has been rewritten as: " The extensive quality control procedures used in this study to minimise errors from the data source and transcription process are an example of the effort required to prepare an observations dataset for analysis."

*P12 line 10-12: Even though the statement is probably generally true, can you conclude that (in such a strong for) from your work? The fraction of errors detected in your dataset is rather small.*

We have removed the final section of this sentence and rewritten it slightly: " Without a reliable method of digitization and a standard method to assess the quality of sources, the accuracy and usability of the final dataset can be jeopardised."

*P12 line 25-26: I think the recommendation of five-time keying applies rather to specific examples (such as these observation take at sea) than it can be considered as a general rule?*

We have updated this sentence to clarify that the reason five-times keying is used here is because the digitizers are volunteers, rather than because the data are taken at sea: " Citizen science efforts which make use of large numbers of volunteers in fact require a value to be keyed at least three, and up to five, times (Eveleigh et al., 2013)".

*P12 line 29: Delete "and useful".*

We could not find this phrase on the mentioned page and line number, and so have removed "and successful" instead.

*P12 line 34-39: What are the concrete conclusions of these assessments? Does it result in integration or removal of data sources? Correction/prevention of errors e.g. by re-scanning original data sources? Giving more specific instructions of digitizers?*

Good point, we have added the following sentence: " With this information it then becomes possible to provide improved instructions to digitisers, develop better templates and tools for digitization, or even rescan data sources if possible."

*P13 line 1-13: This paragraph is rather unclear and I suggest to restructure it. Suggestion: 1. Our findings in the case of Zagazig do not confirm WMOs recommendation regarding templates. Mention some advantages of using templates too. 2. The case highlights the importance of creating user friendly templates. 3. Recommendation of creating user friendly templates (e.g. include feedback of digitizers when designing templates).*

Many thanks for this thoughtful suggestion, we have updated this section accordingly.

*P13 line 22-23: Delete "It is no use… already complete.".*

Done

*P13 line 23-25: Suggestion for this sentence: "Conducting QC as soon as data become available means that the digitizers may detect own keying errors, and that you can advise them on how to increase the quality of their work.".*

We're not sure that asking digitzers to QC their own work is really the best strategy, as they are likely to miss their own mistakes. Instead we have updated this sentence to

read: " While we don't suggest digitizers should QC their own work, conducting QC as soon as data become available by making use of the digitization team means you can advise the digitizers of their errors and hopefully increase the quality future work."

*P13 line 26-31: Did you check for and detect such systematic quality issues? If yes, how was that included in your study respectively why not? If you did not check: Do you think such systematic quality issues are a potential problem in your dataset?*

These errors were largely identified by SAQC. We have added a sentence in this section to clarify: "Flagging and manually examining these errors, as we have done with SAQC, allows all of the affected observations to be retained by one simple correction."

*P14 line 6: "Manual checking of values and decisions based on expert knowledge may mean…"*

Updated.

*P14 line 11: Replace "The digitized dataset" by "It".*

This suggestion has been included in the larger update of this section.

*Table 2: I suggest to always use the same number of decimals within one field.*

Thanks for this suggestion we have updated a number of the wind speed conversions accordingly.

*Table 3: Do not use the same illustration for different tests (Climatic outliers, Big jumps and sharp spikes).*

We have provided a different illustration for the climatic outliers test:

[Figure]

*Table 3: I suggest to mention the change in the Duplicate values test only in the text and not in the table.*

Good suggestion, we have updated the table.

*Table 3: Precipitation totals seems not to be a test but just a daily aggregation. Please specify or remove it from the table.*

We have clarified this to show that the precipitation test flags all values when sum of sub-daily RR data does not equal daily RR total.

*Table 4: Define more clearly to which test the flags relate to (e.g. does lf10 mean "Passed SAQC"?).*

The flags don't relate to a particular test. All tests are applied, and in some cases a value is identified by many tests. Manual interpretation of the test results then determines which flag should be applied.

*Table 5: I think this table could be moved to the supplementary material. Furthermore, only a few tests are spatial consistency test (HQC is described as QC for spatial consistency in the text).*

Thanks for this suggestion, we have moved Table 5 and made it Table S2.

*Table 6: The caption of the table is unclear, please reformulate.*
*Table 6: Observing times are mostly not in accordance with observing times from Table 7. Please explain these differences.*

We have updated the table column names to be clearer, and reformulated the table caption: " The networks used in the spatial and automatic quality control analysis (HQC), including the period, variables and observing times examined. Note that not all observing times were examined in HQC due to neighbouring data availability. " Table 7 provides information on the number of observations per day, not the observation times.

*Figure 1: Remove small titles above figures.*
*Figure 1: Years shown in small titles above figures (1960-2010) not in accordance with time period in legend (1950-2010).*

This figure has been revised, improving the colour scale, removing the small titles and clarifying the period covered by the plots (1957-2010).

*Figure 2: Delete last sentence (same to find in the text) or correct it and provide a direct link to the images.*

We have added "(see section 6)" to this caption, rather than providing the full link again.

*Figure 6: Is the title needed?*
*Figure 6: I also suggest to replace "Percentage of tested data %" by "Flagged observations [%]".*
*Figure 6: Show variable acronyms horizontally.*

Figure updated

*Figure 7: Correct last sentence in caption.*

Done

*Figure 7: I suggest to remove longitude and latitude here.*

We've decided to keep the latitude and longitude for readers who may be unfamiliar with European map projections.

*Figure 7: An indication (or additional plot) of the length of the near-complete time series of these stations after filling data gaps within your study would be an interesting asset for data users.*

We are showing the length of the data series in colours on this plot. While this does show the 'near complete' time series, it does give an indication of long-term records, which we hope will help future data users.

*Figure 8: remove "1e6" from plots.*
*Figure 8 d): Invert scale of hours of the day, separate "daily" more clearly from the rest would be helpful.*
*Figure 8: I suggest to replace "Data count" by "Observations".*

We have incorporated the majority of these suggestions, although we have kept the order of the hourly scale as we can't see what improvement that would make.

*Figure 9: I would remove the first pie-chart from the figure, describing the numbers in the text is sufficient.*

We have replaced Figure 9 with a bar chart rather than a pie chart, and reduced the size of the first dataset. However, we feel that it is valuable to show the number of flagged values relative to the full dataset.

*Figure 9: It is not clear how you chose the flags shown here (% of each flag of all flagged values? Other flags too small to be represented? Very similar to Fig. 12?). Please clarify or remove the figure.*

This figure shows the results of the SAQC procedure only.

*Figure 10 d): I do not think this figure provides much information, I suggest to remove it.*
*Figure 10: Remove "1e1" from figures.*
*Figure 10: I suggest to replace "Percentage of total data %" by "Flagged observations [%]".*
*Figure 10: Colors are hard to distinguish.*

Based on this comment and the suggestion of reviewer 1 we have updated this figure to make the colours clearer, updated the axis labels as suggested and removed 1e1.

*Figure 11: Remove title.*
*Figure 11: Replace "Percentage of tested data %" by "Flagged observations [%]".*

Done.

*Figure 12: Changing % between "Total" and "Flags" is unclear. I suggest to show the fraction of flagged observation of the total in ‰.*
*Figure 12: The order in "Flags" seems a bit random.*
*Figure 12: The color code is unclear. There are three darkness classes for each color, but in the caption only two classes are explained.*
*Figure 12: What group is "Passed SAQC but recheck required"? Why would you recheck observations that passed the SAQC test?*

We have updated the caption of this figure to clarify what is meant by the three shades of darkness, and 'passed SAQC but recheck required. We believe that updating the caption makes the figure clearer, and so have not updated the image.

Update Figure 12 caption: The percentage distribution of quality control flags in the UERRA dataset. Values that have passed QC are represented in green (QC flags fl10, fl40 and fl30); values that were flagged but verified as correct are shown in purple

(fl14, fl44 and fl34); values that were flagged but corrected are shown in blue (fl12 fl42, fl32); and values that were flagged and removed are shown in orange (fl11, fl13, fl15, fl17 and fl36). The darkness of the colors indicates the level of QC applied for each flag. Lighter colors represent values that were only subjected to semi-automatic quality control (SAQC, fl codes that begin with 1), darker colors indicate values subjected to both SAQC and spatial HQC procedures (fl codes that begin with 3), and the colors in the middle represent the small number of values that may need to be rechecked (fl codes that begin with 4). See Table 4 for additional flag details.